# Investigations of Middle-Caliber Anti-Aircraft Cannon Interior Ballistics including Heat Transfer Problem in Estimation of Critical Burst Length

**Bartosz Fikus \***[iD]**, Alicja Dorochowicz, Zbigniew Surma** [iD]**, Jacek Kijewski, Zbigniew Leciejewski,** [iD] **Jakub Michalski** [iD] **and Radosław Trębiński**

Faculty of Mechatronics, Armament and Aerospace, Military University of Technology, 2 Sylwestra Kaliskiego Street, 00-908 Warsaw, Poland; dorochowicz.ala@gmail.com (A.D.); zbigniew.surma@wat.edu.pl (Z.S.); jacek.kijewski@wat.edu.pl (J.K.); zbigniew.leciejewski@wat.edu.pl (Z.L.); jakub.michalski@wat.edu.pl (J.M.); radoslaw.trebinski@wat.edu.pl (R.T.)

\* Correspondence: bartosz.fikus@wat.edu.pl

**Abstract:** Numerical and experimental investigations of armament systems are an important part of modern design processes. The presented paper reports problems that were encountered on the theoretical analysis of the performance of 35 mm anti-aircraft cannon and the way in which they were solved. The first problem concerns the application of results of closed vessel tests of used propellant in interior ballistics simulations. The use of a nonstandard form of the gas generation rate equation solved this problem. The second problem concerned the assessment of projectile–barrel interaction. The barrel resistance was estimated making use of finite element analysis. The third problem arose from the need to determine the heat transfer from propellant gases to the barrel. The employed formula for the heat exchange coefficient and 2D modelling of the heat conduction in the barrel provided the solution. Selected elements of the theoretical model were validated by shooting range experiments and data provided by the ammunition producer. Using the considered approach, crucial ballistic parameters (maximum propellant gas pressure and muzzle velocity) were estimated with an error of less than 6.0%, without application of additional fitting coefficients. The numerical estimation of the barrel external surface temperature provided a relative discrepancy with the experimental data lower than 6% and enabled the estimation of the critical burst length, equal to 14 shots.

**Keywords:** anti-aircraft cannon; interior ballistics modelling; barrel resistance; numerical simulations; heat transfer in barrel

## 1. Introduction

Numerical and experimental investigations of interior ballistics phenomena are an important part of modern armament design and the modernization process [1,2]. Results of these works provide the set of data, necessary in mechanical and thermal examinations of the under-investigation construction. Taking into account the limited number of reports considering the middle-caliber gun (in the literature small arms and large-caliber guns, e.g., 120 mm, were mainly considered), the aim of this paper is to present the problems that the authors encountered on the theoretical analysis of the performance of 35 mm anti-aircraft cannon and the way in which they were solved. The first problem was connected with the characteristics of single-base propellant applied in the under-consideration launching system. To obtain realistic values of these characteristics closed vessel tests were performed. Their results enabled us to apply a nonstandard form of the gas production equation for the interior ballistics simulations. In the simulations, the thermodynamic lumped-parameter model of interior ballistics was applied. Many papers provide data which confirm correctness and efficiency of such a modeling way (e.g., [3–7]).

The second problem concerned the assessment of the interaction between the projectile and the barrel. In the classical approach [3] the proportionality between the projectile

kinetic energy and the resistance work is assumed. This problem for artillery systems was investigated theoretically more thoroughly by several scientific teams, e.g., [7–10]. Authors of [7] investigated numerically the influence of bore wear on the course of barrel resistance force. In this model, the discussed force was included in the fictionally increased projectile mass. As the result of FEA simulations, the influence of barrel bore length and land height were estimated. It was concluded in [8] that, due to the large number of parameters impacting on the barrel resistance force course, it is necessary to conduct simulations of the projectile–barrel interaction for each investigated system. In paper [9] the authors investigated the influence of the implemented computational method on the results of simulations of the rotating band engraving process. All three considered methods, i.e., Lagrangian finite element approach (FEM), meshless method (FEM-SPH), and Lagrangian–Eulerian approach (CEL), provided similar results. Authors of [10] investigated the influence of propellant charge mass on the barrel resistance force. As stated, the pressure course has significant influence on the value of the investigated force. Moreover, the qualitative course of resistance is similarly independent from the propellant charge mass. Similar conclusions can be found in [11], where this effect was explained mainly by the Poisson effect. All cited works showed that for the realistic assessment of the resistance force finite element simulations are necessary. This way was chosen in this paper.

The third problem relates to the theoretical assessment of the critical burst length. Calculations of the temperature distribution in the barrel are necessary for this assessment. A relevant formula for calculation of the heat exchange coefficient value was chosen and 2D/3D heat conduction simulations were performed.

To validate the accepted solutions of the mentioned problems, experimental shooting tests were performed. The producer's ammunition data and measurements of the projectile muzzle velocity allowed for validation of the interior ballistics model and the method of assessment of the resistance force. Application of thermography provided data for estimation of the thermal model correctness.

## 2. Experimental Investigations

### 2.1. Propellant Characteristics

Experimental investigations of propellant characteristics were based on classical closed vessel tests [12]. Propellant gas pressure courses were provided making use of a 200 cm$^3$ closed vessel HPI B180T (HPI, Austria) presented in Figure 1. The gas pressure value was measured using an HPI 5QP6000M piezoelectric transducer (HPI, Austria), characterized by a maximum measurement error of 1%.

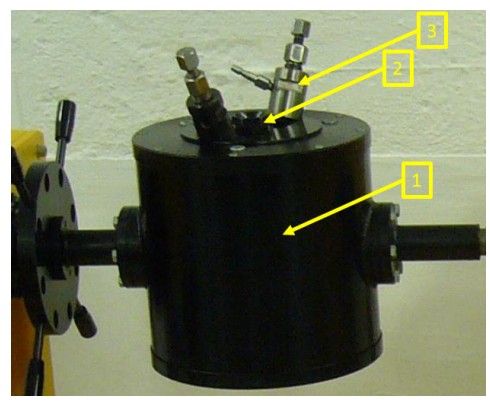

(**a**)

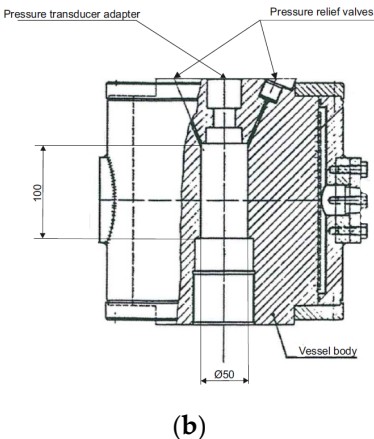

(**b**)

**Figure 1.** Closed vessel HPI B180T: (**a**—appearance; **b**—cross-section (1—closed vessel with jacket, 2—pressure transducer adapter, 3—pressure relief valve).

The considered propellant was a single-base one and its grains (shown in Figure 2) were characterized by dimensions summarized in Table 1.

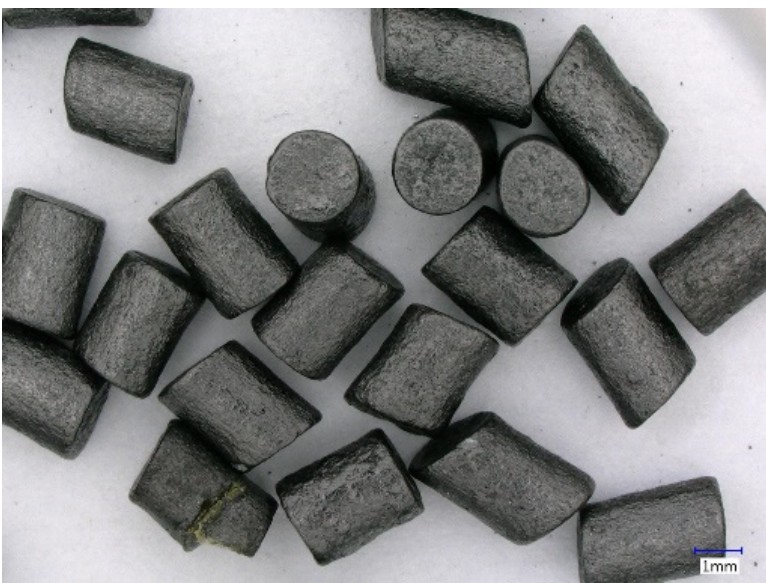

**Figure 2.** Grains shapes of investigated propellant.

**Table 1.** Propellant grain characteristics.

| Parameter | Value |
|---|---|
| Grain type | single-perforated |
| External diameter [mm] | 2 |
| Perforation diameter [mm] | 0.15 |
| Grain length [mm] | 2.8 |
| Web thickness [mm] | 0.925 |

In order to estimate the required characteristics, tests were conducted in conditions of two values of loading density, i.e., $100 \text{ kg/m}^3$ and $200 \text{ kg/m}^3$. For each condition, two tests were carried out. Pressure courses provided by these experiments are presented in Figure 3.

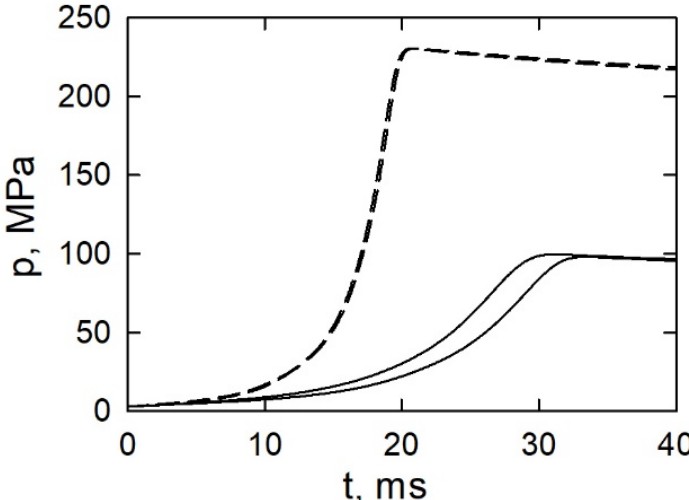

**Figure 3.** Propellant gases pressure courses obtained for loading density of $100 \text{ kg/m}^3$ (solid line) and $200 \text{ kg/m}^3$ (dashed line).

Using the methodology of heat losses correction described in [12,13], where losses are estimated using the descending part of the pressure curve to assess the heat transfer coefficient, the corrected values and courses were obtained. To correct the pressure curve, the values of correction for each measured value should be calculated:

$$\Delta p(t) = \frac{1}{t_r} \int_{t_{ign}}^{t} p(\tau) \, d\tau \tag{1}$$

where $p$ is pressure, $t$—time, $t_r$—resultant characteristic time of pressure decrease, $t_{ign}$—ignition time of propellant bed. The applied $t_r$ constant can be assessed using the following formula:

$$t_r = \frac{\int_{t_{max}}^{t_d} p(t) \, dt}{p_{max} - p_d} \tag{2}$$

where $p_{max}$ is the maximum pressure, $p_d$ is a certain value on the descending part of the pressure curve (assumed value was equal to 0.6 $p_{max}$), $t_{max}$ and $t_d$ denote the time corresponding to the mentioned values of pressure.

The obtained maximum values of pressure (for the gas density characteristic for a completely burnt propellant) enabled assessment of the equation of state (EOS) coefficients using the least-squares approximation. In the presented considerations, the EOS, whose source is the Noble–Abel equation, was applied [3,4]:

$$p_g(\rho_g) = \frac{R_g T_g \rho_g}{1 - \alpha \rho_g} \tag{3}$$

where $R_g$ is the individual gas constant, $T_g$—gas temperature, $\rho_g$—gas density, $\alpha$—co-volume coefficient. The estimated values of the EOS coefficient are summarized in Table 2. The letters E and Q denote the values obtained from uncorrected (E) and corrected for heat loss experimental data. In further investigations, the second set of data was applied.

**Table 2.** Estimated equation of state parameters for propellant gases.

| Parameter | Value |
|---|---|
| (E) $f = R_g T_{g0}$ [kJ/kg] | 826 |
| (Q) $f = R_g T_{g0}$ [kJ/kg] | 895 |
| (E) $\alpha$, [dm$^3$/kg] | 1.366 |
| (Q) $\alpha$, [dm$^3$/kg] | 1.153 |

In the above-presented table, $T_{g0}$ means the flame temperature of burning propellant. Assuming the following burning law:

$$\frac{dz}{dt} = G(z) p_{atm} \left( \frac{p_g}{p_{atm}} \right)^n \tag{4}$$

where $z$ denotes the relative burnt mass of propellant, $t$ is time, $G(z)$ is the dynamic vivacity function, $p_{atm}$ is the atmospheric pressure, $p_g$ is the propellant gases pressure and $n$ is the law exponent; it is possible to estimate the dynamic vivacity function course and the burning law exponent [13]. In the presented paper this was performed making use of the manipulated Equation (4):

$$\log_{10}(\frac{dz}{dt}) = \log_{10}(G(z) p_{atm}) + n \log \left( \frac{p_g}{p_{atm}} \right) \tag{5}$$

Using values of $dz/dt$ and $p_g$ for discrete values of $z$ in the interval between 0.3 and 0.8, it is possible to estimate pressure exponent value $n$ making use of the linear

regression for each value of $z$. The estimated average value of $n$ was equal to 0.961. The dynamic vivacity course as the function of relative burnt propellant mass was evaluated using Equation (4) and is presented in Figure 4. The first period of the burning process is seriously disturbed by different course of the ignition process in closed vessel conditions in comparison with ammunition. To minimalize this influence, the first segment of the $G(z)$, curve was approximated by linear function (red line).

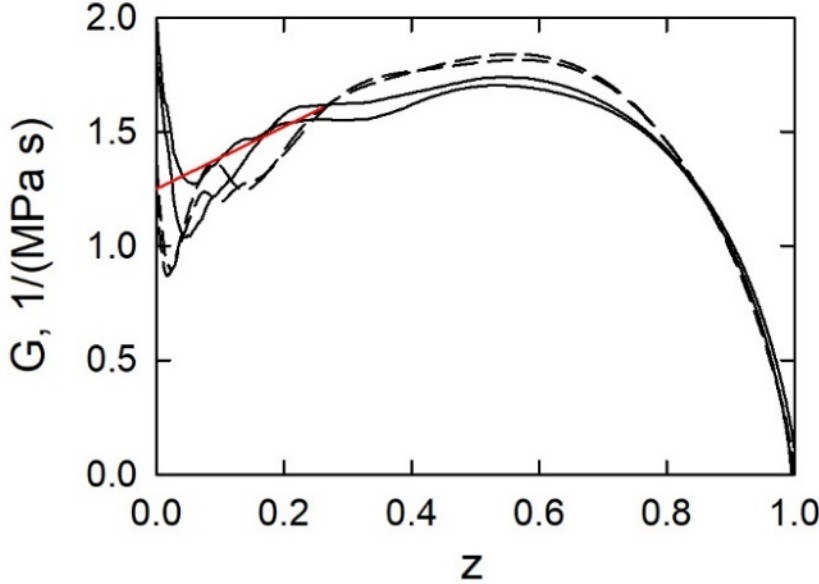

**Figure 4.** Estimated dynamic vivacity function as the function of relative burning propellant mass for loading density of 100 kg/m$^3$ (solid line) and 200 kg/m$^3$ (dashed line).

The values of $dz/dt$ as a function of $z$ and $p_g$ can be calculated using the relations:

$$dz/dt = (dz/dp_g)(dp_g/dt) \tag{6}$$

$$z = \frac{b_1 p_s}{f + b_2 p_s}, \quad p_s = p_g - p_{ign}, \quad b_1 = \frac{1}{\Delta} - \frac{1}{\rho_g}, \quad b_2 = \alpha - \frac{1}{\rho_g} \tag{7}$$

where $p_{ign}$ is the pressure generated by the ignition system and $\Delta$ is the loading density.

Pressure time derivative values are calculated based on the recorded pressure courses.

### 2.2. Ballistics Characteristics

Ballistics characteristics needed for validation of the theoretical model were determined during in-field shooting. Test were carried out using the measurement set presented in Figure 5. Applied Doppler radar Weibel SL-525PE (Weibel, Lillerød, Denmark) and the high-speed camera Phantom v1612 (Vision Research, Wayne, NJ, USA) allowed for measurements of muzzle velocity, which was estimated based on 7 rounds. Doppler radar was located at the cannon. In accordance with the producer's data and known discrepancy between muzzle velocity and the value extrapolated to the muzzle using Doppler radar data (from external ballistics region), the maximum error of the muzzle velocity estimation can be equal to 1%, which results in approximately 10 m/s overestimation for the considered velocity range. In order to minimize the measurement error, the high-speed camera was positioned 15 meters from the muzzle (perpendicular to the barrel axis). Additional application of the thermographic camera FLIR E60 (FLIR, Wilsonville, ON, USA) was used to measure the temperature increase of the external barrel wall surface. Supplementary tests conducted in laboratory conditions allowed for estimation of temperature measurement accuracy of the applied approach. Comparison with values registered by the K-type thermocouple provided maximum discrepancy with the thermographic method

equal to 2 °C for the under-investigation temperature range (15–50 °C). For all registered values, the thermographic results were overestimated, but the temperature differences were characterized by a lower value of uncertainty. Field measurements were carried out in a burst regime of fire, i.e., 6-round bursts from a distance of 3 m. To improve the correctness of measurements, a linear gauge was applied in the FLIR software. For each burst, the resultant temperature increase was estimated, which enabled the estimation of the mean temperature changes for each shot. To provide data for the validation of the theoretical model, the temperature increase of the selected barrel region (Figure 6) was estimated.

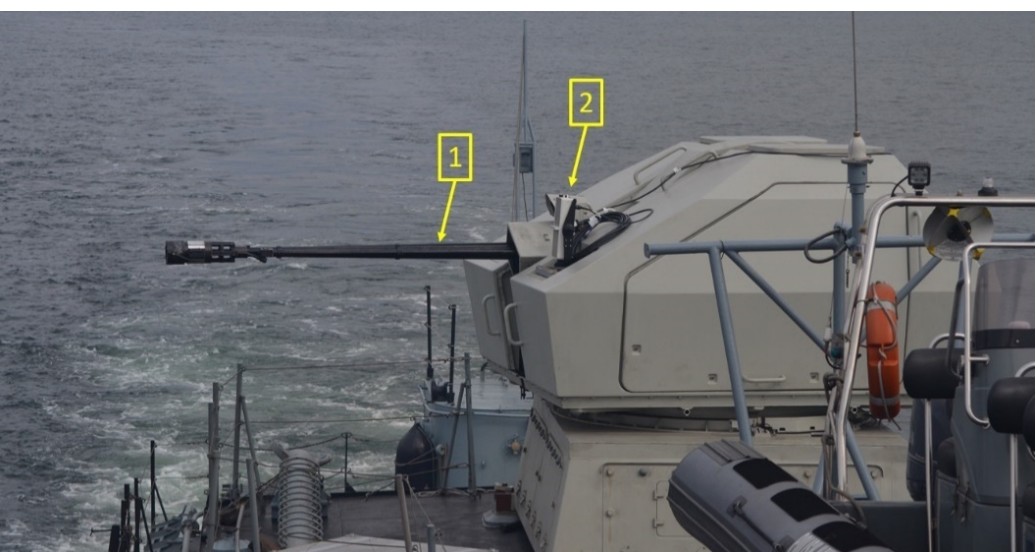

**Figure 5.** Investigated launching system (1) with Doppler radar (2).

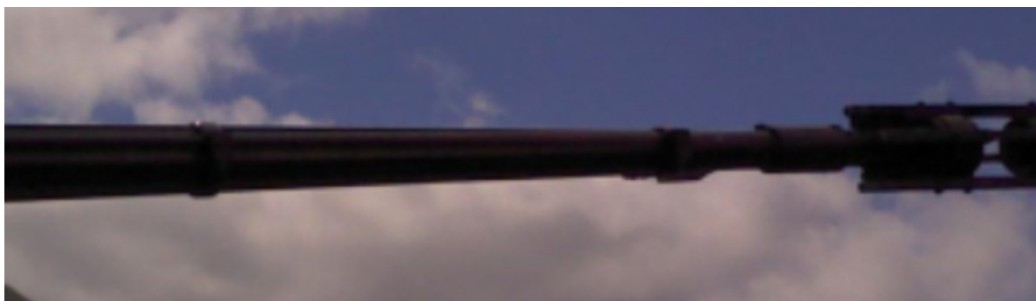

**Figure 6.** Barrel region investigated using the thermographic camera.

The conducted experimental investigations provided data of the projectile muzzle velocity. Results of measurements conducted using the high-speed camera and Doppler radar are summarized in Table 3. The results obtained using the camera confirm the correctness of the radar measurements, which were applied in the model validation.

**Table 3.** Results of velocity measurements.

| No. of Shots | Value of Projectile Velocity [m/s] | |
|---|---|---|
| | **High Speed Camera** | **Doppler Radar \*** |
| 1 | 1172 | 1175 |
| 2 | 1170 | 1170 |
| 3 | 1175 | 1180 |
| 4 | 1170 | 1178 |
| 5 | 1174 | 1171 |
| 6 | 1173 | 1182 |
| 7 | 1162 | 1160 |
| average | 1170.9 | 1173.7 |
| standard deviation | 4.34 | 7.50 |
| max–min | 13 | 22 |

\* Measurements of Doppler radar were applied in the further model validation. The high-speed camera measurements were applied to verify the radar results in case of a disturbed radar signal.

After the experimental investigations of the projectile velocity for single shots, measurements of the increase of the barrel temperature for the burst fire were carried out. Exemplary temperature distribution on the barrel external surface is presented in Figure 7. The values of changes of temperature for the applied gauge are summarized in Table 4.

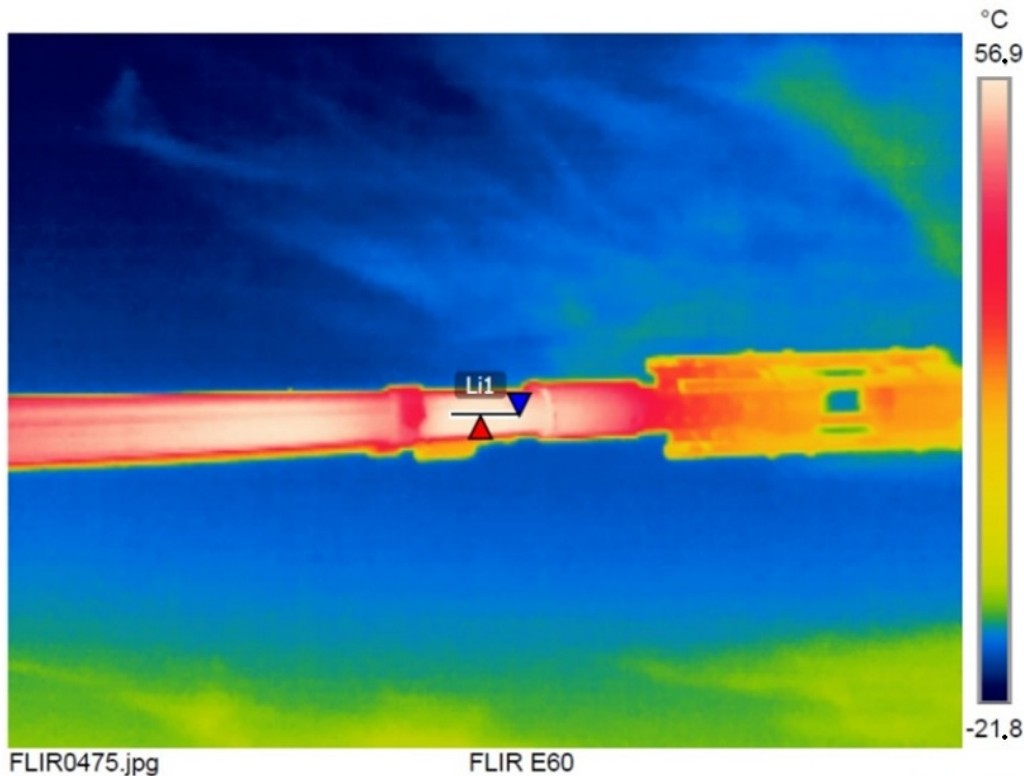

**Figure 7.** Temperature distribution of the barrel external surface in the near-muzzle region.

**Table 4.** Results of temperature measurements.

| No. of Shots in Burst | Value of Temperature [deg. C] | | Temperature Increase [deg. C] |
|---|---|---|---|
| | Before Burst | After Burst | |
| 6 | 20.4 | 58.0 | 37.6 |
| 6 | 29.3 | 78.9 | 49.6 |
| 6 | 60.4 | 101.4 | 41.0 |
| average temperature increase for burst | | | 42.7 |
| max–min | | | 12.0 |

## 3. Numerical Simulations

Considered in this paper model of interior ballistics is the lumped-parameters model [14], based on the thermodynamic approach to modelling interior ballistics phenomena. In comparison with the classical approach, presented in [3], the model includes the explicit form of secondary works carried out by propellant gases. This fact forced the necessity to estimate the barrel resistance force, which was assessed in a numerical way.

### 3.1. Interior Ballistics Model

The model is represented by the set of ordinary differential equations, expressing the fundamental conservation laws:

- Projectile trajectory equation:

$$v_p = \frac{dl_p}{dt} \tag{8}$$

where $v_p$ is the projectile velocity, $l_p$ is its displacement, $t$ stands for time.

- Projectile equation of motion [4]:

$$\frac{dv_p}{dt} = \frac{(p_p - p_{br} - p_{air})s_p}{m_p} \tag{9}$$

where $p_p$ is the propellant gas pressure acting on the projectile base, $p_{br}$ is the barrel resistance pressure, $p_{air}$ is the pressure of air compressed in front of the projectile, $s_p$ is the projectile cross-section area and $m_p$ is the projectile mass.

Due to the existence of gas pressure gradient, the value of pressure acting on the projectile base was estimated making use of the following relation [4]:

$$p_p = p_g + \left( \frac{\omega(p_{br} + p_{air})}{3m_p} \right) \Big/ \left( 1 + \frac{\omega}{3m_p} \right) \tag{10}$$

where $p_g$ is the average propellant gas pressure and $\omega$ is the propellant mass.

The pressure of air in front of the projectile, $p_{air}$, was estimated using the following relation [14]:

$$p_{air} = p_{atm} + v_p^2 \rho_0 \frac{\gamma_{air} + 1}{4} + \sqrt{v_p^4 \rho_0^2 \left( \frac{\gamma_{air} + 1}{4} \right)^2 + v_p^2 \gamma_{air} \rho_0 p_{atm}} \tag{11}$$

where $\rho_0$ is the initial air density and $\gamma_{air}$ is the air heat capacity ratio.

- Propellant gases generation rate equation (4).
- Equation defining the gas temperature changes rate [14]:

$$\frac{dT_g}{dt} = \frac{\omega \frac{dz}{dt}(q_{pow} - c_{vg}T_g) + (c_{vg}\omega + c_{vair}m_{air})T_g\frac{d\xi}{dt} - \frac{dW_{sum}}{dt} - \frac{dH_{out}}{dt}}{c_{vg}\omega(z - \xi) + c_{vair}m_{air}(1 - \xi)} \tag{12}$$

where $T_g$ is the propellant gases temperature, $\omega$ is the propellant mass, $q_{pow}$ is the isochoric heat of combustion, $c_{vg}$ denotes the specific heat of propellant gases at constant volume, $W_{sum}$ is the total work made by gases, $\xi$ is the relative mass of outflowed gases, $m_{air}$ is the mass of air initially present in the case, $c_{vair}$ is the air specific heat at constant volume, $H_{out}$ is the enthalpy of outflowing gases. Moreover, the following differential equations describing the enthalpy of outflowing gases were applied [14]:

$$\frac{dH_{out}}{dt} = \left(c_{pg}\omega + c_{pair}m_{air}\right)T_g\frac{d\xi}{dt} \tag{13}$$

where $c_{pg}$ and $c_{pair}$ are the specific heat of the propellant gases and air at constant pressure, respectively. The rate of change of the total work made by gases was expressed by the following differential equation:

$$\frac{dW_{sum}}{dt} = \frac{dE_{kin}}{dt} + \frac{dW_{br}}{dt} + \frac{dW_{air}}{dt} + \frac{dW_{term}}{dt} \tag{14}$$

where $E_{kin}$ is the total kinetic energy of projectile and propellant-gas mixture, $W_{br}$ is the work done against barrel resistance, $W_{air}$ is the work done against the pressure of air in front of the projectile, $W_{term}$ is the heat losses. The above-described change rates can be estimated by the following formulae [3,4,14]:

$$\frac{dE_{kin}}{dt} = \left(I_p\frac{4\pi^2}{\eta^2} + m_p + \frac{\omega}{3}\right)v_p\frac{dv_p}{dt} \tag{15}$$

$$\frac{dW_{br}}{dt} = s_p p_{br} v_p \tag{16}$$

$$\frac{dW_{air}}{dt} = s_p p_{air} v_p \tag{17}$$

$$\frac{dW_{term}}{dt} = \int_{s\,int} h_{term}\left(T_g - T_{bs}\right)ds \tag{18}$$

where $I_p$ is the projectile moment of inertia, $\eta$ denotes the rifling twist, $s_{int}$ is the heat exchange surface, $h_{term}$ is the heat transfer coefficient, $T_{bs}$ is the barrel internal wall temperature. The heat transfer coefficient was estimated using the form of approximation available in the literature for pipe interior flows [15,16]:

$$h_{term}(x,t) = 0.023\text{Re}^{0.8}\text{Pr}^{0.3} \tag{19}$$

$$\text{Re} = \frac{\rho_{gas}v_{gas}d_b}{\mu_{gas}} \tag{20}$$

where $x$ is the axial coordinate, $d_b$ denotes barrel internal diameter, $\mu_{gas}$ is the dynamic viscosity coefficient of propellant gases, $v_{gas}$ is the propellant gases velocity.

Coefficients of the above-mentioned expression were estimated making use of the approximate propellant gases composition, summarized in Table 5 [17,18]. Diffusive transport coefficients were assessed using the molar fraction weighted averaging of the temperature functions of the dynamic viscosity and thermal conductivity of pure species. Estimated functions are presented in Figure 8 [18].

**Table 5.** Approximate propellant gas composition.

| Compound | Mass Fraction [%] | Molar Fraction [%] |
|----------|-------------------|--------------------|
| $CO_2$ | 17.4 | 9.4 |
| CO | 53.0 | 44.9 |
| $H_2O$ | 15.8 | 20.8 |
| $H_2$ | 1.2 | 14.2 |
| $N_2$ | 12.6 | 10.7 |

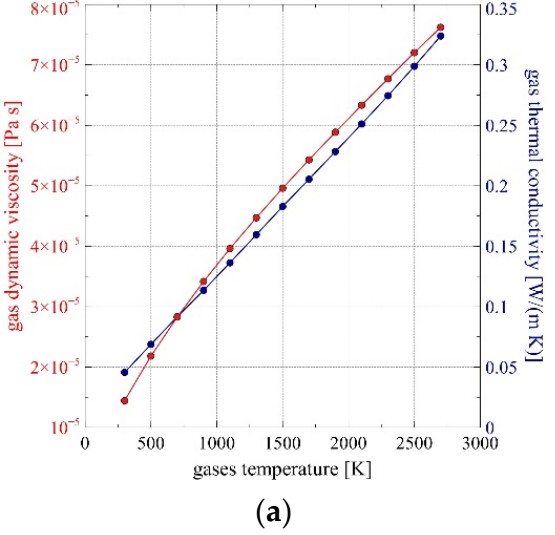

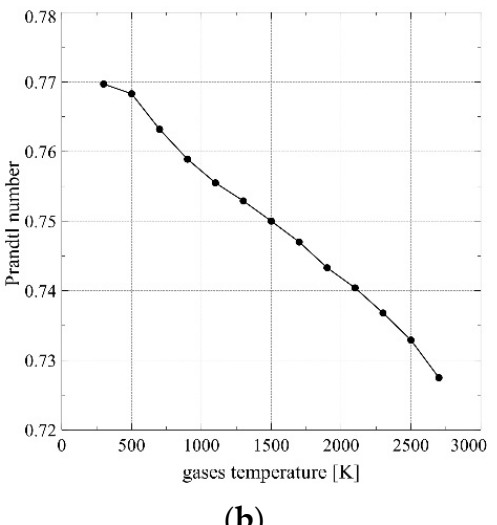

**(a)**       **(b)**

**Figure 8.** Diffusive transport coefficients (a) and Prandtl number (b) of propellant gases as a function of temperature.

In order to include the internal barrel wall temperature changes, the two-dimensional Fourier–Kirchhoff equation was solved:

$$\frac{\partial T_b(x,r,t)}{\partial t} = \frac{1}{c_b \rho_b}\left(\frac{1}{r}\frac{\partial}{\partial r}\left(\lambda_b r \frac{\partial T}{\partial r}\right) + \frac{\partial}{\partial x}\left(\lambda_b \frac{\partial T}{\partial x}\right)\right) \tag{21}$$

where $T_b$ is the barrel material temperature, $r$ is the radial coordinate, $\lambda_b$ denotes the barrel material thermal conductivity, $c_b$ is the specific heat and $\rho_b$ denotes the barrel material density.

The above-presented equation was supplemented by the condition of initial temperature (300 K) and the following boundary condition on the internal barrel surface:

$$\left(\vec{n}_{surf} \cdot \nabla T_b\right)_{surf} = -\frac{h_{term}}{\lambda_b}\left(T_{bs} - T_g\right) \tag{22}$$

where $\vec{n}_{surf}$ in the vector normal to the surface.

- Propellant gases outflow equation [3,14]:

$$\frac{d\xi(t)}{dt} = \frac{s_p}{\omega + m_{air}}\left(\frac{2}{\gamma_g + 1}\right)^{\frac{1}{\gamma_g - 1}}\sqrt{\frac{2\gamma_g}{\gamma_g + 1}}\frac{p_g}{\sqrt{R_g T_g}} \tag{23}$$

where $\gamma_g$ is the propellant gases heat capacity ratio.

- Equation of state in form (1). To include the influence of the multicomponent nature of the mixture, Dalton's law was applied. In the case of air, the perfect gas equation of

state was assumed (i.e., $\alpha = 0$). The propellant gases density was estimated using the following relation [14]:

$$\rho_g = \frac{\omega \cdot z}{W_0 - \frac{\omega}{\delta}(1 - z)} \tag{24}$$

The above-presented system of equations allows for estimation of the crucial ballistic parameters of the barrel launching system, i.e., time courses of the gas pressure and projectile velocity.

### 3.2. Projectile–Barrel Interaction Model

As mentioned in the previous subsection, the interior ballistics model requires the value of the barrel resistance force. Finite element analysis was used for determining it. The geometry of the TPT ammunition produced by MESKO (Poland) was taken into account. The basic data of the under-investigation system are summarized in Table 6.

To conduct numerical simulations, the CAD geometry of the real model (Figure 9a) was simplified and meshed (Figure 9b). Geometry simplifications included barrel shortening to 1200 mm bore (to ensure an acceptable simulation time) and reduction of chamfers (to ensure satisfactory mesh quality). The applied mesh consisted mainly of hexahedral elements. Considering the results of [9], the classical Lagrangian FEM formulation was applied.

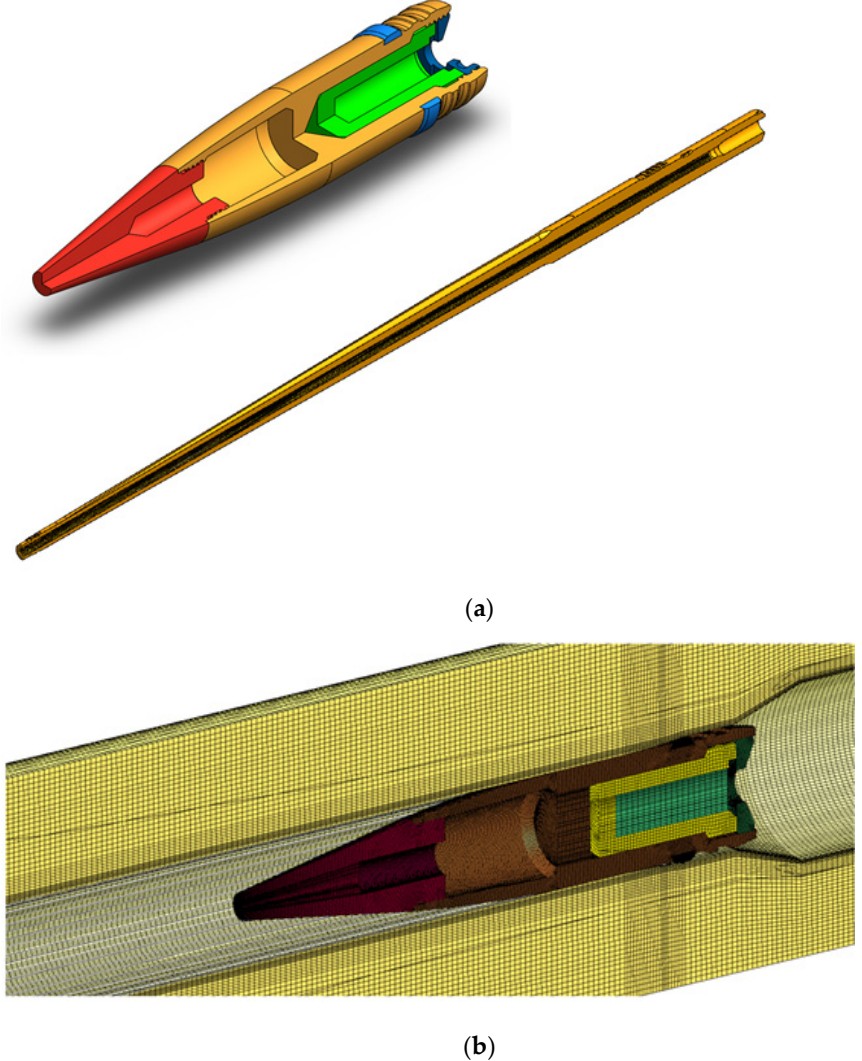

(a)

(b)

**Figure 9.** Geometry of the under-investigation system ((**a**)—exact geometry; (**b**)—simplified and meshed geometry).

In the presented investigations, the barrel was made of steel, which was assumed to be elastic-perfectly-plastic. Similar assumptions were made for the projectile body—made of steel and aluminum alloy. The crucial element, i.e., the rotating band was made of OFHC copper, which was assumed to be elastic-plastic material. In order to take into account the high strain rate and thermal effects the Johnson–Cook model was used to assess the yield stress of the material [19]:

$$Y = (A + B\varepsilon^n)\left(1 + C\ln\dot{\varepsilon}^*\right)(1 - T^{*m}) \tag{25}$$

$$T^* = \frac{T - T_0}{T_m - T_0}, \ \dot{\varepsilon}^* = \frac{\dot{\varepsilon}}{\dot{\varepsilon}_0} \tag{26}$$

where $A, B, C, n,$ and $m$ denote model parameters, $\varepsilon$ is the strain, $T$ is the material temperature.

**Table 6.** Basic data of the under-investigation system [20].

| Parameter | Value |
|---|---|
| Barrel caliber [mm] | 35 |
| Barrel length [mm] | 3150 |
| Rifling twist [deg] | linearly variable from 0 to 6.5 |
| Projectile mass [g] | ~550 |
| Projectile muzzle velocity obtained using ballistic barrel [m/s] | $1180 \pm 15$ |
| Average maximum gas pressure estimated based on series of shots [MPa] | $\leq 420$ |

Furthermore, the Johnson–Cook failure model was applied. In this case the failure parameter is estimated by the following expression [19,21]:

$$D = \sum \Delta D = \sum \frac{\Delta\varepsilon_{pl}}{\varepsilon_f^{JC}} \tag{27}$$

where $\Delta\varepsilon_{pl}$ is the effective plastic strain increment. The effective plastic strain at failure is estimated by:

$$\varepsilon_f^{JC} = [D_1 + D_2\exp(D_3\sigma^*)]\left(1 + D_4\ln\dot{\varepsilon}^*\right)(1 + D_5T^*) \tag{28}$$

where $D_1, D_2, D_3, D_4$ are the model parameters, $\sigma^*$ is the stress triaxiality.

In the presented model, the following main boundary conditions were applied:

- fixed barrel inlet;
- gas pressure acting on the projectile bottom;
- all parts of the projectile tied;
- contact between the rotating band and the barrel imposed with a penalty-based formulation including erosion of the failed elements.

As the initial condition, the initial velocity of all parts was assumed to be equal to zero.

For a projectile displacement greater than 1200 mm (i.e., after the engraving process, for medium and low pressure acting on the projectile bottom), the barrel resistance force was extrapolated proportionally to the pressure acting on the projectile bottom [11]. The proportionality factor was estimated satisfying the barrel resistance force continuity.

### 3.3. 3-Dimensional Heat Transfer Model

To provide data for validation of the heat exchange model, FE simulations of heat transfer in the second half (near muzzle region) of the barrel were conducted. The aim of this part of the calculations, was the estimation of the barrel external wall surface temperature increase as a function of time. The reason for commercial code application (Ansys for meshing and LS-DYNA for heat transfer problem simulations) is the generally complicated

shape of the barrel. The applied design has an important impact on the thermal capacity of the barrel. Moreover, the commercial code allowed for verification of the heat transfer problem solution algorithm, applied in the interior ballistics model. Due to the quasi-axisymmetric shape of the barrel in the second-half area it was also possible to consider the 2D model for verification purposes. During the simulations, the FE models presented in Figure 10 were investigated. In the case of the 3D model, the symmetry of the investigated thermal problem allowed for application of a quarter of the full geometry, ensuring a shorter time for computations. Similar numerical investigations of the 2D heat transfer problem for the 35 mm barrel launching system were considered in papers [22,23]. Due to the unphysical model formulation, i.e., short and intensive rectangular approximation of internal surface thermal loading, the authors of the mentioned papers obtained unrealistic results (barrel internal surface temperature of 2200 K, which exceeds the material melting temperature).

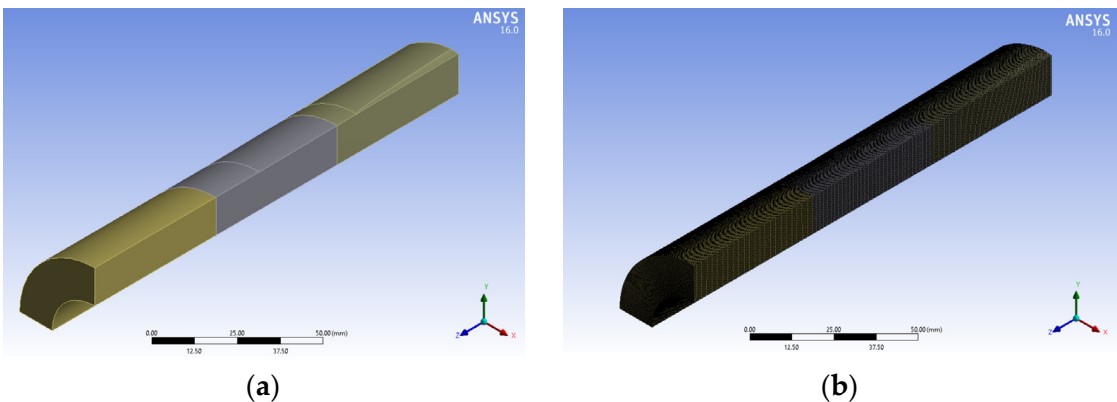

(**a**)                                                                (**b**)

**Figure 10.** Geometry (**a**) and mesh (**b**) for the under-investigation heat transfer problem.

*3.4. Model Parameters and Results of Numerical Simulations*

Numerical simulations of the under-investigation phenomena were conducted in the iterative way (using the method of successive approximation). Results of the interior ballistics model were implemented as the above-mentioned boundary condition in the FE simulations conducted with LS-DYNA explicit code [24]. This approach, using the projectile equation of motion, allowed for iterative estimation of the barrel resistance force and, using the interior ballistics model, the courses of gas pressure and projectile velocity as a function of projectile displacement and time. The calculation process was conducted for Courant–Friedrichs–Lewy number CFL = 0.7. The mesh sensitivity analysis provided an acceptable value of the rotating band elements dimension, equal to 0.13 mm. Material models parameters applied during simulations were summarized in Tables 7 and 8 [19,21,25,26] and the parameters characterizing the under-investigation launching system are presented in Table 9.

**Table 7.** Constitutive model parameters applied during simulations.

| Parameter | Value | | |
|---|---|---|---|
| **Material** | **OFHC Copper** | **Steel** | **Aluminum Alloy** |
| Material density [kg/m$^3$] | 8960 | 7850 | 2710 |
| Young's modulus [GPa] | 124 | 210 | 69 |
| Poisson's ratio | 0.34 | 0.30 | 0.30 |
| Specific heat [J/(kg·K)] | 383 | | |
| Shear modulus [GPa] | 45 | | |
| Yield strength [MPa] | | 830 | |
| Melting temperature [K] | 1356 | | |
| Initial (room) temperature [K] | 300 | | |
| Constant $A$ [MPa] | 90 | | |
| Constant $B$ [MPa] | 292 | | |
| Constant $C$ [-] | 0.025 | | |
| Exponent $n$ [-] | 0.31 | | |
| Exponent $m$ [-] | 1.09 | | |
| Reference strain rate $\dot{\varepsilon}_0$ [s$^{-1}$] | 1 | | |

**Table 8.** Johnson–Cook failure model parameters for OFHC copper.

| Parameter | $D_1$ | $D_2$ | $D_3$ | $D_4$ | $D_5$ |
|---|---|---|---|---|---|
| **Value [-]** | 0.540 | 4.889 | −3.030 | 0.014 | 1.120 |

**Table 9.** Launching system parameters applied in simulations.

| Parameter | Value |
|---|---|
| Initial chamber volume $W_0$ [dm$^3$] | 0.360 |
| Projectile displacement to the muzzle $l_m$ [mm] | 2930 |
| Projectile mass $m_p$ [kg] | 0.550 |
| Propellant mass $\omega$ [kg] | 0.345 |
| Propellant "force" $f$ [kJ/kg] | 895 |
| Co-volume coefficient $\alpha$ [dm$^3$/kg] | 1.153 |
| Burning law exponent $n$ [-] | 0.961 |
| Propellant gases specific heat ratio $\gamma_g$ [-] | 1.2 |
| Propellant heat of combustion $q_{pow}$ [MJ/kg] | 4.48 |
| Propellant density $\delta$ [kg/m$^3$] | 1550 |
| Gas constant of propellant gases $R_g$ [J/kg·K] | 350 |
| Gas constant of air $R_{air}$ [J/kg·K] | 287 |
| Isochoric specific heat of propellant gases $c_{vg}$ [J/kg·K] | 1750 |
| Isochoric specific heat of air $c_{v\,air}$ [J/kg·K] | 750 |
| Primer pressure $p_{ign}$ [MPa] | 7 |

In the case of the 2D self-developed and 3D commercial codes heat transfer model, the temperature dependent material properties summarized in Table 10 were applied. Due to the availability of limited data at this stage of the investigations, the simplified simple approximations of thermal conductivity and specific heat (Figure 11) were applied [22,23,27,28]. During the numerical simulations, taking into account the high rate of temperature changes, aa temperature increase limit during one time step was imposed. The computations were carried out for the time sufficient to reach the maximum temperature at the external barrel surface. Mesh size sensitivity analysis conducted for single shot allowed for the assumption of the element size applied during simulations. Results of the influence of mesh size on the interior barrel surface maximum temperature (obtained during preliminary numerical tests), which is the most sensitive on grid dimension, are presented in Figure 12. The estimated optimal value was equal to 0.0375 mm and the applied elements were hexahedral. During the simulations the implicit scheme was applied (diagonal scaled conjugate gradient iterative solver).

**Table 10.** Barrel material thermophysical properties and values of temperature of processes.

| Parameter | Value |
|---|---|
| Material density [kg/m$^3$] | 7850 |
| Thermal conductivity [W/(m·K)] | 19 + 0.014(T-293) |
| Hardening temperature [K] | 1200–1250 |
| Tempering temperature [K] | 800–930 |

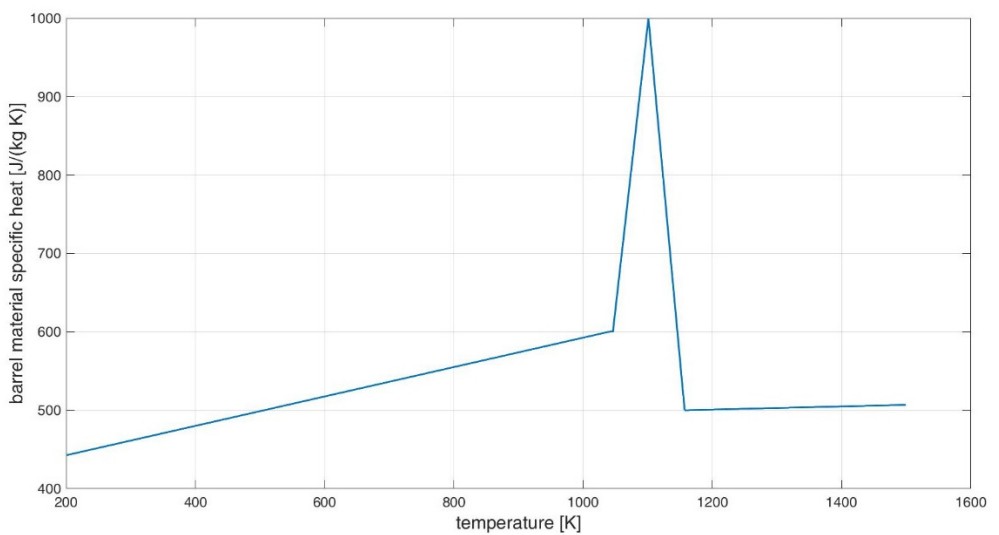

**Figure 11.** Approximation of barrel material specific heat as a function of temperature.

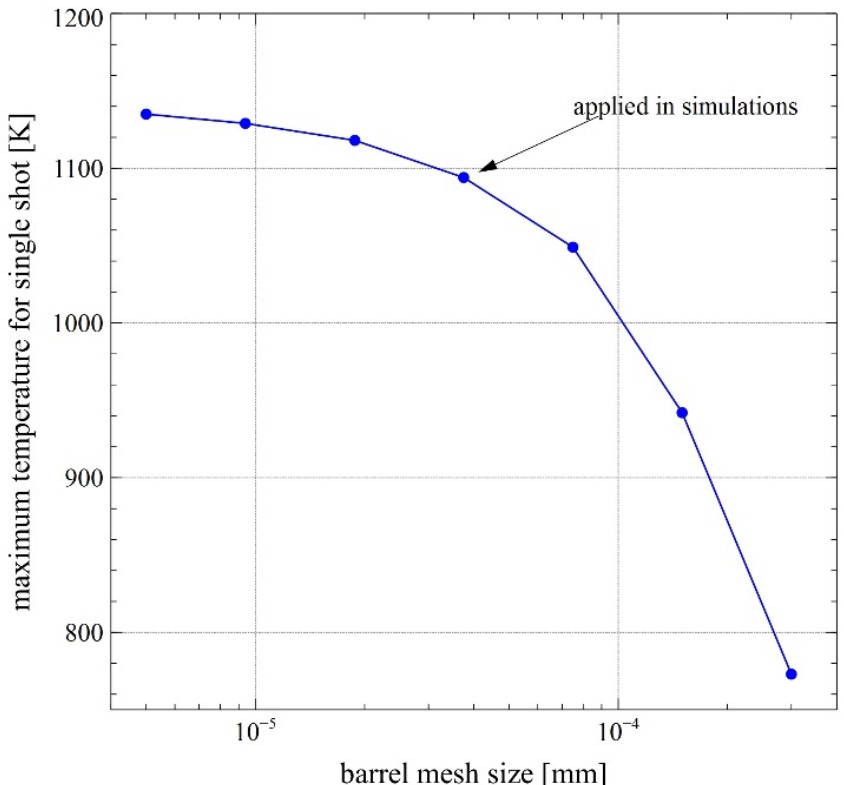

**Figure 12.** Influence of barrel mesh size on the maximum temperature of the barrel interior surface for single shot.

The results of numerical considerations, the ballistic curves (courses of projectile velocity and gas pressure as the function of time) were estimated and are presented in Figure 13. As can be observed, for so high a propellant mass, it is necessary to include the propellant gas pressure gradient. It is of note, that the obtained maximum pressure obtained in simulations (395 MPa) is acceptably close to the producer's data mentioned in Table 6 (420 MPa), ensuring approx. 6.0% of discrepancy. The muzzle velocity provided by simulations was equal to 1129 m/s, giving 3.8% discrepancy with the experimental value without application of fitting coefficients.

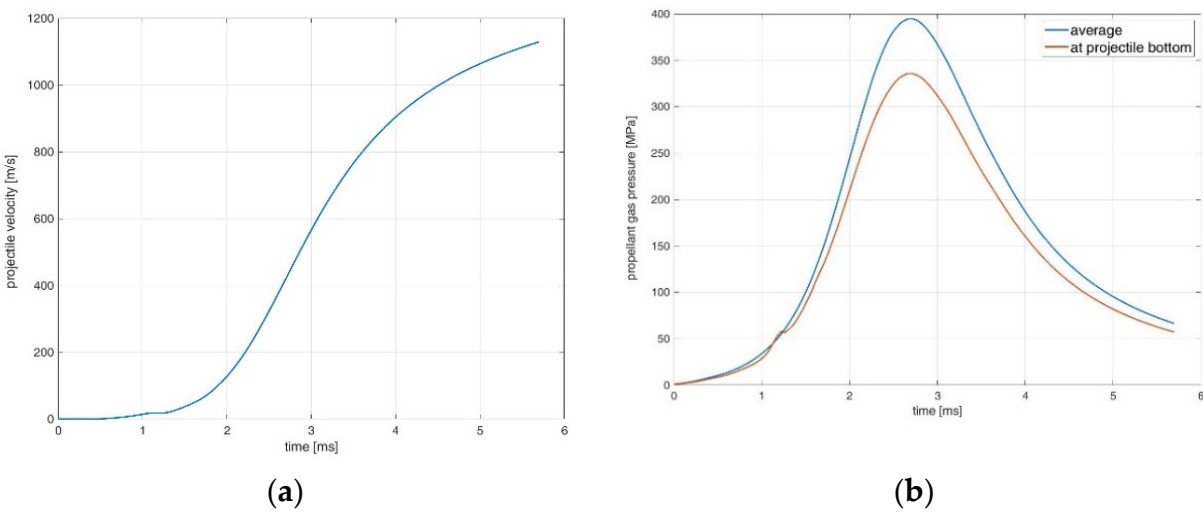

(**a**)         (**b**)

**Figure 13.** Ballistic curves (as function of time), (**a**)—projectile velocity, (**b**)—propellant gas pressure.

Moreover, the barrel resistance pressure (defined as the resistant force divided by the barrel cross-section area) as the function of projectile displacement was estimated and is presented in Figure 14. Two important extrema of considered pressure can be noticed there. The first one (approx. 72 MPa) corresponds to the deformation of the projectile body threshold (presented in Figure 15). The second extremum (approx. 55 MPa) is the result of the maximum projectile acceleration generated by the maximum value of propellant gas pressure [11].

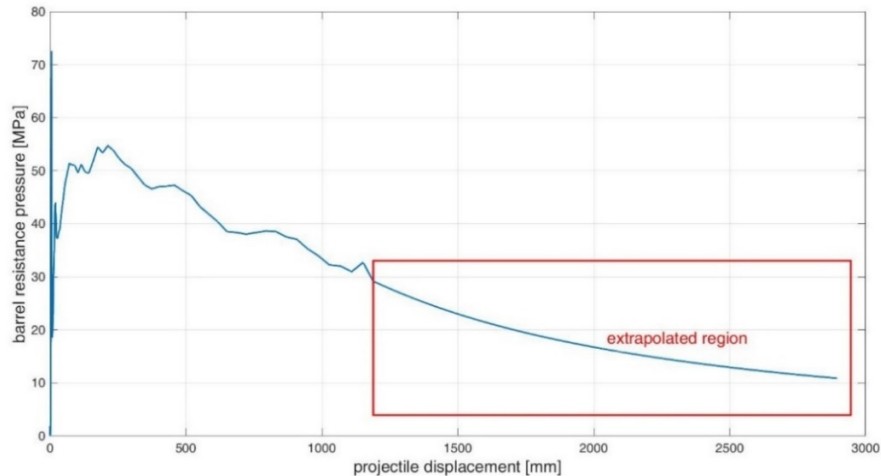

**Figure 14.** Barrel resistance pressure as function of projectile displacement.

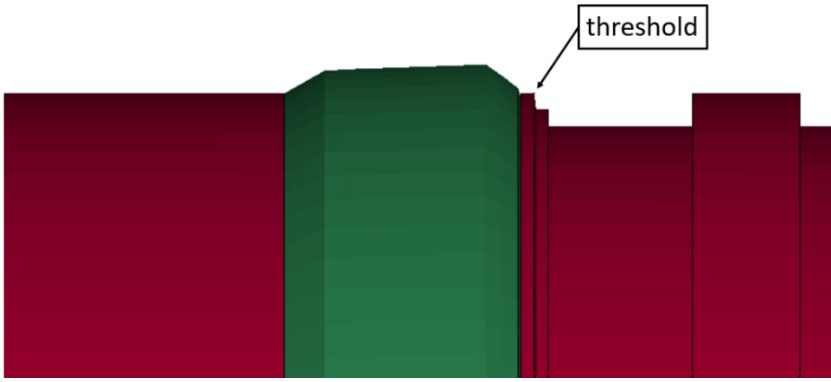

**Figure 15.** Shape of rotating band with projectile body threshold.

Coupling of mechanical and thermal problems enabled estimation of the barrel internal surface heat loading for two cross-sections. The first one corresponds to the cross-section which was investigated using a thermographic technique and characterized by the lowest heat capacity. On the other hand, the second one is the most thermally loaded cross-section close to the chamber and characterized by the largest heat capacity due to l the large barrel wall thickness. The courses of heat flux at the considered surfaces for the 6-round burst are presented in Figure 16. As can be seen, the gas-barrel heat transfer intensity decreases for each shot, which is the result of the barrel surface temperature increase during firing. The observed transferred heat value reduction coefficient (relative to the first shot), defined using the following formula:

$$r_n = \frac{\int\limits_{time\ of\ shot\ n} h_{term}(T_g - T_{bs})dt}{\int\limits_{time\ of\ shot\ 1} h_{term}(T_g - T_{bs})dt} \tag{29}$$

can be treated as high and equal to 0.91 for the second shot and 0.76 for the sixth shot. These high values are the result of relatively long intervals between shots (0.11 s) and the high internal surface cooling rate generated by significant heat flux divergence in the barrel wall.

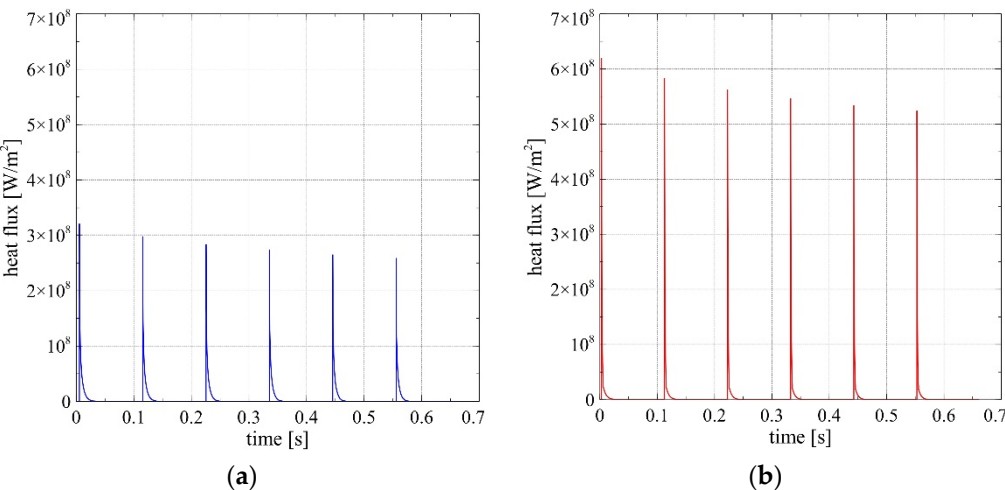

**Figure 16.** Time courses of the heat flux at the internal barrel surface ((**a**)—for the validation cross-section of the barrel; (**b**)—for the most loaded cross section).

As the main result of the thermal model simulations, the temperature courses for the internal surfaces for the first and second cross-sections were estimated and are presented in Figure 17. As it can be observed, for each considered shot in the case of the most loaded

cross-section, the maximum material temperature exceeds the phase transition temperature. The observed rapid cooling of the material ensured by heat transport in the barrel allows for hardening of the material, without significant changes of material properties relative to its initial conditions. On the other hand, for the second cross-section, the phase transition temperature was not reached in the considered case, but the cooling process was comparably fast relative to the first investigated cross-section. The temperature course for the external barrel surface of the first considered cross-section applied in the model validation is presented in Figure 18. The slowly-changing character of the temperature increasing process (temperature raising time equal to 23 s), gives the basis for estimation of the sampling rate during further experimental investigations with temperature registration.

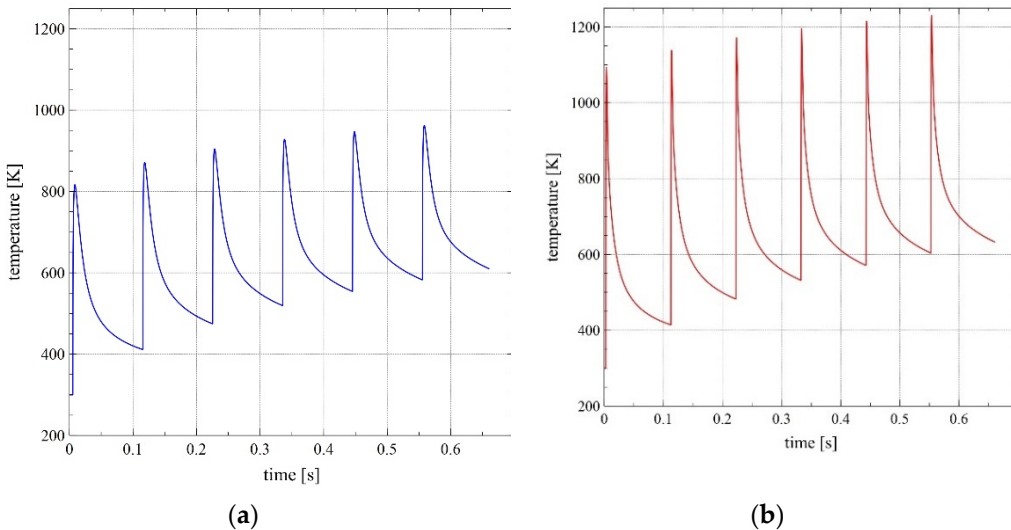

(**a**)                                          (**b**)

**Figure 17.** Time courses of the temperature at the internal barrel surface for validation cross-section (**a**) and the most loaded cross-section (**b**).

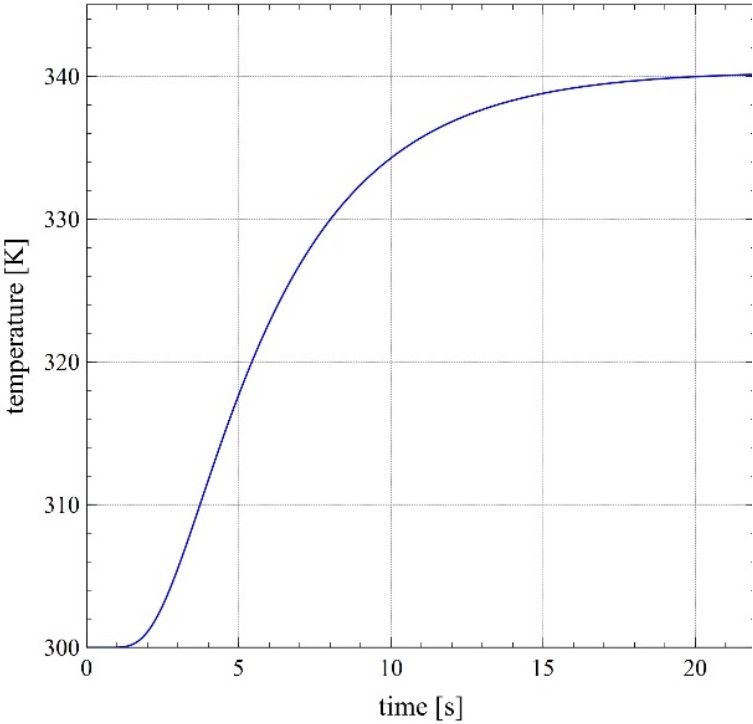

**Figure 18.** Time courses of the temperature at the external barrel surface at the validation cross-section.

The provided maximum values of the external barrel surface temperature seem to be real and are equal for the under-consideration barrel region—40.2 K. Moreover, two considered approaches (2D and 3D) provided approximately the same values of temperature changes (the noticed discrepancy in peak value was equal to approx. 2%), which positively verified both models. The obtained order of magnitude for temperature increase are comparable with the results obtained by other researchers, who investigated artillery systems, e.g., [28–31].

The results of the transient thermal analyses were necessary for estimation of the critical burst length. To estimate this limit, we can formulate and apply two main criteria. First, when the temperature introduces a serious decrease of material strength, it can be observed above 670–770 K (400–500 deg. C [32]). Taking into account this statement, the critical burst length is the number of shots which produces a temperature of 770 K in the barrel material at the beginning of the next shot, which can overload the material due to the propellant gas pressure.

The second formulated criterion is based on material transitions and defines the critical length as the number of shots which generates the temperature of the upper limit of the tempering process (930 K) in the whole investigated barrel cross-section after burst. The considered conditions would ensure a sufficiently slow cooling rate to temper the under-investigation material. The above mentioned temperature and the tempering process would introduce changes in comparison with the process applied by the producer. Taking into account the minimum of the above-mentioned values, the first criterion should be applied in the under-consideration problem.

The considered element, i.e., the critical burst length estimation, is important in the case of the anti-aircraft middle-caliber cannon due to its high fire rate and low mass, which is not observed in the case of large-caliber guns (e.g., howitzers or tank guns, which normally shoot several rounds per minute).

## 4. Discussion

The applied numerical model of the interior ballistics phenomena seems to provide realistic results. The obtained maximal value of propellant gas pressure corresponds with the producer's data, providing 6% of relative discrepancy independently of the applied heat transfer coefficient definition. A similar discrepancy with experimental data is noticeable in values of muzzle velocity. Comparison of experimentally obtained data with the results of numerical simulations shows underestimation of the estimated values, providing 4.8% relative discrepancy (56 m/s) for the applied assumptions.

The first reason for the observed discrepancies can be associated with the modelling of the propellant burning process. The dynamic vivacity curve shown in Figure 4 is determined not only by the geometry of the propellant grains but also by the ignition process of the propellant bed. This process in the closed vessel differs form that in the case chamber.

The second reason for the discrepancies is associated with the modeling of the rotating band–barrel interaction. The implemented model does not include the band wearing process, which would decrease the barrel resistance force for the post-maximum period of shot and provide a higher value of muzzle velocity. Moreover, barrel resistance extrapolation could additionally introduce some error. In accordance with the literature, for moderate values of propellant pressure, the used extrapolation can be applied [11], but it is only a rough estimation of the under-consideration force.

Worth noticing is the significant value of the barrel resistance. The classical approach, described in [3], assumes proportionality between the kinetic energy of the projectile and the resistance work. The same refers to the heat losses. Plots shown in Figure 19, based on the results of simulations, prove, that these assumptions are very far from the real conditions. Therefore, the barrel resistance and the heat losses should be included in the explicit forms, as was done in this paper. The implemented iterative approach simplifies the simulation process. The final results were obtained after the second iteration of the FEA

calculations. Moreover, it was observed, that for first FEA iteration the barrel material can be assumed to be rigid, which significantly reduces the computational cost of the whole process without noticeable differences in the obtained results.

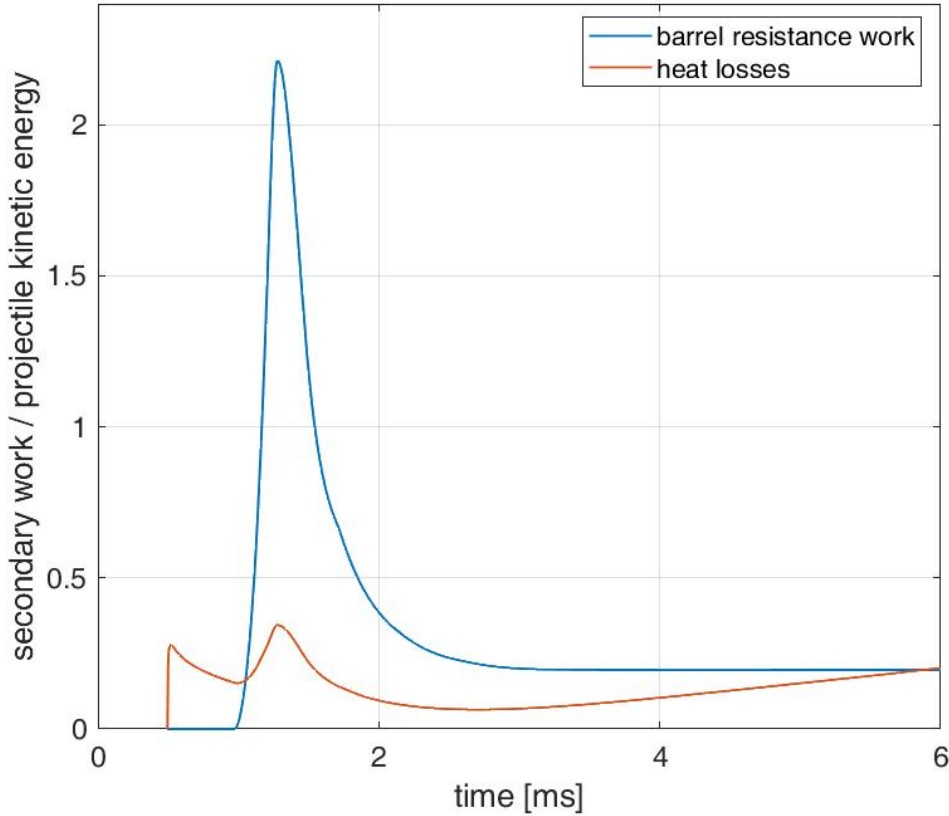

**Figure 19.** The courses of the relation of secondary works to the projectile kinetic energy as function of time.

Numerical estimation of the barrel temperature increase ensured that the results agreed well with the experimental data. Application of expression (14) resulted in only 6% underestimation of temperature increase (in comparison with the mean value). Estimated values of the temperature increase were included in the experimental data dispersion interval. Moreover, the estimated high values of the transferred heat reduction coefficient (24), suggest the possibility of application of the heat flux estimated for a single shot in the thermal analyses for only very short bursts (e.g., three shots) in the case of similar medium-caliber launching systems.

One of the most important points of the presented paper, i.e., estimation of the critical burst length, was carried out making use of the iterative process. Conducted simulations for long bursts allowed for the assessment of the critical length, which was equal to approx. 14 shots. Results of calculations of the radial distribution of the barrel material temperature for the most loaded region in the cases of 14- and 20-shot bursts are presented in Figure 20. As can be noticed for the 14-shot burst, the narrow layer (0.2 mm) at the barrel interior surface starts to reach a temperature above 770 K. The estimated number of shots (especially the 20-round series for which the overheated layer is characterized by a thickness of 1.2 mm) can be dangerous for the barrel construction and can intensify the barrel wearing process. The obtained value of the critical burst corresponds with the recommendations of the producer. The time gap should be applied after 15-shots for an intensive fire regime to allow for temperature equalization.

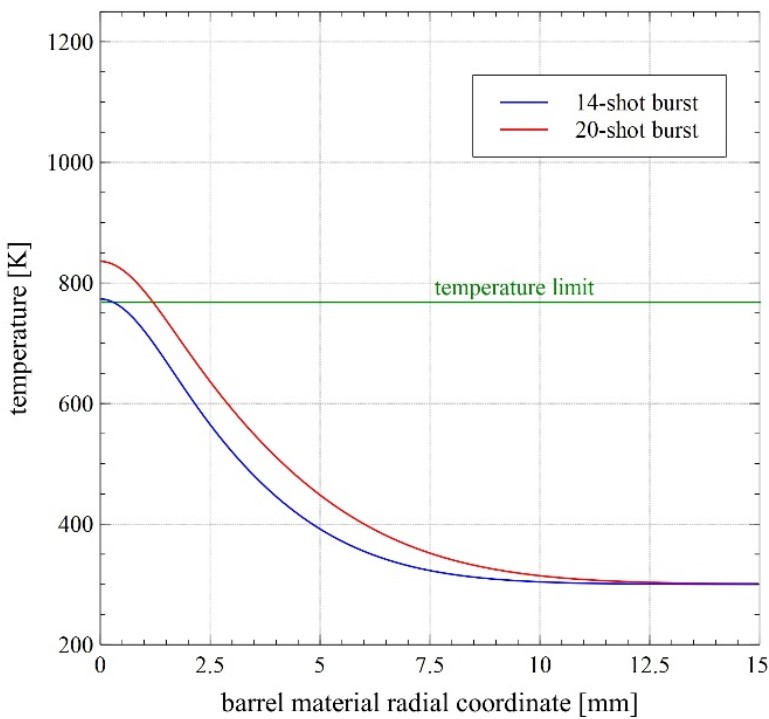

**Figure 20.** Barrel material temperature radial distribution.

Considering the possible influence of the barrel external surface cooling conditions on results of measurements (e.g., wind etc.) it is reasonable to investigate the influence of the external heat transfer coefficient on the results of the calculations. The dependence of this parameters on the external surface temperature increase is presented in Figure 21. As expected, the influence of the considered parameter is relatively low and can be treated as less than 3% with respect to the initially assumed value for the extremely high heat transfer coefficient (50 W/m$^2$K). Moreover, the estimated value is not dependent on the applied internal heat flux definition.

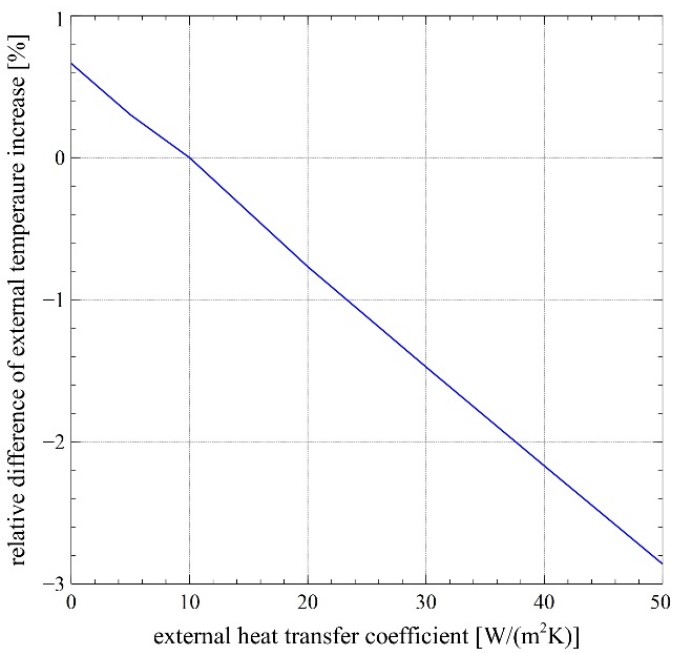

**Figure 21.** The influence of external surface heat transfer coefficient on the temperature increase of the external surface.

## 5. Conclusions

The conducted investigations provided the following conclusions:

- data obtained directly from closed vessel tests enable modeling of the interior ballistics problems for artillery systems (due to relatively coarse propellant grains), providing sufficient accuracy of the theoretical results;
- interior ballistics models should include the barrel resistance force in the explicit form [4]. The interaction process is extended, and it is not possible to approximate it using only the start pressure and the modified projectile mass [3];
- as a novelty, we can conclude, that the applied iterative process of barrel resistance estimation and involving it in a numerical model (hybrid approach) seems to provide an acceptable force estimation without fully-coupled models;
- the theoretical estimation of barrel temperature increase (using simplified expressions defining heat flux between gases and barrel surface) provided acceptable discrepancy with the experimental data and can be recommended in similar analyses;
- heat transfer between the propellant gases and the barrel wall is one of the most important losses and it is necessary to include this effect in simulations of interior ballistics of artillery (even middle caliber) systems;
- the conducted analyses enabled estimation of the critical burst length, equal to ca. 14 shots, which agrees with the producer's recommendations. In our opinion, the fire regime proposed by the producer should not be changed.

**Author Contributions:** Conceptualization, B.F.; methodology, B.F., R.T. and A.D.; software, B.F., A.D. and R.T.; validation, B.F., A.D., J.M. and J.K.; formal analysis, B.F. and R.T.; investigation, B.F., A.D., Z.S., J.K., J.M. and R.T.; resources, Z.L.; data curation, B.F., Z.S., J.K., J.M. and R.T.; writing—original draft preparation, B.F.; writing—review and editing, A.D. and R.T.; project administration, Z.L.; funding acquisition, Z.L. All authors have read and agreed to the published version of the manuscript.

**Funding:** This research was funded by The National Centre for Research and Development, grant number O ROB 0046 03 001 (DOBR/0046/R/ID1/2012/03). The APC was funded by grant number O ROB 0046 03 001 (DOBR/0046/R/ID1/2012/03).

**Institutional Review Board Statement:** Not applicable.

**Informed Consent Statement:** Not applicable.

**Data Availability Statement:** Not applicable.

**Acknowledgments:** Authors would like to thank Judyta Sienkiewicz, Dawid Goździk and Damian Szupieńko for their assistance during experimental investigations.

**Conflicts of Interest:** The authors declare no conflict of interest.

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
