# Peer review of "Investigations of Middle-Caliber Anti-Aircraft Cannon Interior Ballistics including Heat Transfer Problem in Estimation of Critical Burst Length"

_processes, doi:10.3390/pr10030607_

Round 1
Reviewer 1 Report
The main goal of the manuscript is not clear. The authors wanted to provide results of theoretical and experimental investigations of 35 mm anti-aircraft cannon. What kinds of the results did they want to present?
- The abstract should be rewritten. It is not easy to understand the main goal of the manuscript from the abstract.
- After some paper reviews, the authors should gave the readers some summaries about the references.
- The results shown in experimental investigations section must be described. The relations between Fig. 3 and Table 2 are not clear. How to get the results of Fig. 4?
- The equations used in numerical simulations section should be cited.
- In Figure 15, the scales should be the same for a and b.
- The explanations about Figs. 16 and 17 should be rewritten.
Author Response
Dear Reviewer
Thank you very much for your time, suggestions and the reviews. We have read them very carefully and we have corrected our manuscript according to comments of all Reviewers. Therefore, we hope that the new version of manuscript is better than the previous one.
Please see the attachment.
On behalf of authors of the manuscript
Bartosz Fikus

Reviewer 2 Report
The submitted manuscript describes the investigation of middle-caliber anti-aircraft cannon interior ballistics including heat transfer problem in estimation of critical burst length using experimental and numerical approaches. The positive qualities of this manuscript are that it is detailed on all sections and it promises further contribution. For improvement of the manuscript, the following points should be addressed properly:
For experimental approach, how to decide experimental condition is still not clear. The information of accuracy during measurement is necessary. This work seems marginal since how to perform the experimentation and the data reduction is also not clear. Uncertainty of all determined parameters is not informed in detail. The experimental work needs the improvement significantly.
In the numerical approach, the description of the governing equations of a very well known approach, so actually a citation to the existing literature would be enough. There is no significant improvement for the numerical method. Reporting results based on a computational study, even if a commercial code (here using commercial software Ansys Fluent) is used, must include a discussion on numerical accuracy. Grid/mesh convergence examined by mesh independency test is necessary. It seems more an application of an existing method, however should be confirmed with the experiment. How to maintain the properties of gas composition (Table 5) and how to ensure without reaction mechanisms occurred during actual process should be declared. How to decide the turbulence model is also important. Some 3D contours is needed to present for providing the easy understanding of physical mechanism.
Moreover, a more comprehensive discussion is required among Figs. 18-20. How to overcome "As can be seen in the Figure 18, this assumption is very far from the real conditions" in Line 466 has to solve clearly in this present work.
Author Response

(The authors gave the same response as above.)

Reviewer 3 Report
The study investigated experimentally and theoretically the propellant properties of a 35 mm anti-aircraft cannon including projectile-barrel interaction and heat transfer problems. The numerical estimation for crucial ballistic parameters and barrel external surface is consistent with the experimental results. Regarding to the presentation and novelty of the study, the following comments are offered.
- The objective of the study should be clarified based on the past literature.
- In figure 1 (b), unit should be offered.
- Definition of co-volume coefficient, α, should be offered.
- In equation 2, how to estimate the n valve of 0.961 and dynamic vivacity function should be clarified.
- In equation 4, the definition of mp should be given.
- The citations from eg. (5) to eq. (13) should be provided.
- For equation (14), why is heat transfer coefficient, hterm, function of x?
- In table 9, how to obtain the launching system parameters in simulations? Are the parameter values consistent with the practical conditions in physics?
- In conclusions, some novelty findings about ballistic phenomenon are suggested to be included.
- Turbine blade heat transfer prediction in flow transition using k-omega two-equation model
Author Response

(The authors gave the same response as above.)

Round 2
Reviewer 1 Report
The authors had rewritten the manuscript and made it easier to be read.
Reviewer 2 Report
This revision version could be accepted.
Reviewer 3 Report
The manuscript has been revised in terms of review comments. It is recommended to be accepted and published in processes.